# Microscopic study of the Halperin–Laughlin interface through matrix product states

V. Crépel [1], N. Claussen[1], N. Regnault[1] & B. Estienne[2]

Interfaces between topologically distinct phases of matter reveal a remarkably rich phenomenology. We study the experimentally relevant interface between a Laughlin phase at filling factor $\nu = 1/3$ and a Halperin 332 phase at filling factor $\nu = 2/5$. Based on our recent construction of chiral topological interfaces (*Nat. Commun.* https://doi.org/10.1038/s41467-019-09168-z; 2019), we study a family of model wavefunctions that captures both the bulk and interface properties. These model wavefunctions are built within the matrix product state framework. The validity of our approach is substantiated through extensive comparisons with exact diagonalization studies. We probe previously unreachable features of the low energy physics of the transition. We provide, amongst other things, the characterization of the interface gapless mode and the identification of the spin and charge excitations in the many-body spectrum. The methods and tools presented are applicable to a broad range of topological interfaces.

[1] Laboratoire de Physique de l'École Normale supérieure, ENS, Université PSL, CNRS, Sorbonne Université, Université Paris Diderot, Sorbonne Paris Cité, Paris 75005, France. [2] Laboratoire de Physique Théorique et Hautes Énergies, LPTHE, Sorbonne Université, CNRS, F-75005 Paris, France. Correspondence and requests for materials should be addressed to V.C. (email: crepel@lpa.ens.fr)

Topological phases of matter, while being gapped in the bulk, often display gapless edge modes. On the one hand the edge modes are controlled by bulk topological invariants, and on the other hand, the critical edge theory governs the full topological content of the bulk. This phenomenon is known as the bulk-edge correspondence (see for instance refs. [1–3] for general discussions about its validity). In the context of the fractional quantum Hall (FQH) effect, the bulk-edge correspondence has been pushed one step further with the pioneering work of Moore and Read[4] who expressed a large class of FQH model WFs as two-dimensional Conformal Field Theory (CFT) correlators. Assuming generalized screening, one can then establish that the associated low-energy edge modes are described by the same CFT[5], making the correspondence between the bulk and edge properties explicit.

Among the challenges that emerged during the last few years in the realm of topologically ordered phases, understanding the interface between two distinct intrinsic topologically ordered phases stands out as one of the most fascinating[6–14]. Predicting what happens at such an interface is notoriously difficult[15–17], and even the transitions between Abelian states have not yet been classified[18–20]. In this article, we consider the prototypical example of an interface between two FQH Abelian states.

The FQH effect is the first observed[21] quantum phase of matter with intrinsic topological order. Many features of this strongly-correlated, non-perturbative problem were unraveled by the study of model wavefunctions (WFs)[4,22]. In particular, the experimentally observed fractional $e/3$ charges[23,24] were first described as excitations of the seminal Laughlin WF at filling factor $\nu = 1/3$[25]. Model WFs have been an invaluable tool in our understanding of the FQH effect. They provide a bridge between the microscopic models in terms of strongly-correlated electrons and the low-energy effective description in terms of topological quantum field theories. Indeed the Laughlin WF is the densest zero energy ground state of a microscopic relevant Hamiltonian, while at the same time it exhibits non-Abelian anyons and a topology-dependent ground state degeneracy consistent with that of a Chern–Simons theory[26,27].

Theoretical approaches to understand interfaces between topological phases mostly rely on the cut and glue approach[12], in which both phases are solely described by their respective edge theories. The interface emerges from the coupling between the two edges[28] and predictions can be made about the nature of the interface theory[7,10,16]. While powerful, these effective field theory approaches suffer from a complete lack of connection with a more physical, microscopic description. In order to overcome this limitation, we have recently proposed a family of matrix product state (MPS) model wavefunctions for the Laughlin–Halperin interface capable of describing the whole system including both bulks and the interface[29]. We found that these MPSs faithfully describe the bulks intrinsic topological order while presenting the expected universal low-energy physics at the interface. However, the validity of these model WFs at the microscopic level still has to be established.

In the present article, we provide such a detailed microscopic analysis of these model WFs through extensive comparison with exact diagonalization (ED). We focus on the fermionic interface between the $\nu = 1/3$ Laughlin state and the $\nu = 2/5$ Halperin (332) state. This interface is relevant for condensed matter experiments, and could be realized in graphene. There, the valley degeneracy leads to a spin singlet state at filling fraction $\nu = 2/5$[30,31] while the system at $\nu = 1/3$ is spontaneously valley-polarized[31–33]. Thus, changing the density through a top gate provides a direct implementation of the Laughlin–Halperin interface.

The paper is organized as follows: we first introduce a microscopic model reproducing the physics of the transition between a Laughlin phase at filling factor 1/3 and a Halperin (332) phase at filling 2/5. We then detail the MPS construction of both the Halperin (332) state, derived in ref. [34], and of the model state for such an interface that we have introduced in ref. [29]. From there we explicitly show how to perform the identification of the interface gapless theory and apply the procedure to our system. We finally compare the ED results of the microscopic Hamiltonian with the model state and show how the construction of the ansatz may be used to identify the spin and charge excitations in the many-body spectrum. The main results of this work appear in this discussion, as it validates the model state not only on the universal features it holds but more importantly at a microscopic level.

## Results

**Microscopic model.** The Halperin $(m, m, m-1)$ at filling factor $\nu = \frac{2}{2m-1}$ is the natural spin singlet[35–37] generalization of the celebrated, spin polarized, Laughlin state[25,38] at filling factor $\nu = 1/m$. It describes a FQH fluid with an internal two-level degree of freedom[39,40] such as spin, valley degeneracy in graphene or layer index in bilayer systems. For the sake of conciseness, we will refer to the internal degree of freedom as *spin* in the following. The most relevant case for condensed matter systems is $m = 3$, namely the fermionic (3, 3, 2) Halperin state. Numerical evidence[30,41] suggests that the plateau at filling $\nu_H = 2/5$ observed in graphene[31] is in a valley-pseudospin unpolarized Halperin (332) state. Interestingly, graphene brought at filling $\nu_L = 1/3$ spontaneously valley-polarizes[31–33] and is described by a Laughlin state. Top-gating different regions to change the density, it seems possible to engineer a setup where the Halperin 332 and Laughlin 1/3 topological orders develop on either side of a sample. These phases have distinct intrinsic topological orders and hence cannot be adiabatically connected to one another. Thus, the creation of such an interface requires a gap closing and the emergence of a critical boundary or of critical points. In this section, we exhibit a microscopic model describing such an interface between the Halperin (332) and the polarized Laughlin 1/3 phases.

We first recall the expressions of the Laughlin 1/3 state

$$\Psi^{\mathrm{Lgh}}(z_1, \cdots, z_{N_e}) = e^{-\frac{1}{4\ell_B^2}\sum_i |z_i|^2} \prod_{1 \leq i < j \leq N_e} \left(z_i - z_j\right)^3, \quad (1)$$

and of the Halperin 332 state

$$\Psi^{\mathrm{Hlp}}(z_1, \cdots, z_{N_e/2}, w_1, \cdots, w_{N_e/2},) = e^{-\frac{1}{4\ell_B^2}\sum_i |z_i|^2 + |w_i|^2}$$
$$\prod_{i<j}\left(z_i - z_j\right)^3 \left(w_i - w_j\right)^3 \prod_{i,j}\left(z_i - w_j\right)^2, \quad (2)$$

where the position of the $i$-th spin down (resp. up) electrons is denoted by $z_i$ (resp. $w_i$) and $\ell_B$ the magnetic length of the system. Here, it is implicit that the total many-body state is the proper antisymmetrization of the spatial part Eq. (2) associated with the spin component $(\downarrow \cdots \downarrow \uparrow \cdots \uparrow)$ with respect to both electronic spin and position. The vanishing properties of these states ensure that they completely screen the interacting Hamiltonian[1,42,43]:

$$\mathcal{H}_{\mathrm{int}} = \int \mathrm{d}^2\mathbf{r} \sum_{\sigma,\sigma' \in \{\uparrow,\downarrow\}} - : \rho_\sigma(\mathbf{r})\nabla^2\rho_{\sigma'}(\mathbf{r}) : + : \rho_\uparrow(\mathbf{r})\rho_\downarrow(\mathbf{r}) : + \mu_\uparrow\rho_\uparrow(\mathbf{r}),$$
$$(3)$$

respectively for $\mu_\uparrow = 0$ and $\mu_\uparrow = \infty$. Here $\mu_\uparrow$ is a chemical potential for the particles with a spin up, $\rho_\sigma$ denotes the density of particles with spin component $\sigma$, and $: \quad :$ stands the for normal ordering. Hence, creating an interface between these two topologically ordered phases can be achieved by making $\mu_\uparrow$ spatially dependent without tuning the interaction[6].

We make a few additional assumptions, allowing the numerical study of such an interface. First, we send the cyclotron energy to infinity, i.e., large enough so that only the Lowest Landau Level (LLL) is populated. We will always assume periodic boundary conditions along the $y$-axis, thus mapping the system on a cylinder with perimeter $L$. Let $c_\sigma^\dagger(\mathbf{r})$ be the creation operator of an electron of spin $\sigma$ at position $\mathbf{r} = (x, y)$. The LLL is spanned by the one-body orbital WFs:

$$\psi_n(\mathbf{r}) = \frac{e^{ik_n y}}{\sqrt{L\sqrt{\pi}}} e^{-\frac{(x-x_n)^2}{2\ell_B^2}}, \tag{4}$$

where the momentum along the compact dimension $k_n = \left(\frac{2\pi}{L}\right)n$, with $n \in \mathbb{Z} + 1/2$, labels the orbitals and determines the center of the Gaussian envelope

$$x_n = k_n \ell_B^2 \tag{5}$$

along the cylinder axis. The corresponding creation operator is $c_{n,\sigma}^\dagger = \int d^2\mathbf{r}\, \psi_n(\mathbf{r}) c_\sigma^\dagger(\mathbf{r})$. Once projected to the LLL, the Hamiltonian of Eq. (3) with a spatially dependent chemical potential $\mu_\uparrow(\mathbf{r})$ is made of an interaction $\mathcal{H}_{\text{int}}$ term and a polarization term $\mathcal{H}_{\text{pol}}$. After projection, the Halperin 332 state (resp. the Laughlin state) becomes the densest zero-energy state for a uniform chemical potential $\mu_\uparrow = 0$ (resp. $\mu_\uparrow = \infty$)[26]. Indeed, $\mathcal{H}_{\text{int}}$ reduces to the zero-th and first Haldane pseudo-potentials[27,44]. We now choose $\mu_\uparrow(\mathbf{r})$ such that the quadratic part reads

$$\mathcal{H}_{\text{pol}} = U \sum_{k<0} c_{k,\uparrow}^\dagger c_{k,\uparrow}, \tag{6}$$

corresponding to a smooth ramp from zero to $U$ in real space over a typical distance $2\pi/L$. Moreover, we assume that $U \gg |\mathcal{H}_{\text{int}}|$ while remaining smaller than the cyclotron energy. This allows us to project our Hilbert space onto a subspace where the occupation for all orbitals with $n < 0$ is zero for the spin up electrons. The polarized Hilbert subspace is spanned by the occupation basis:

$$\left| \left\{ n_k^\downarrow \right\}_k, \left\{ n_k^\uparrow \right\}_{k>0} \right\rangle = \left| \cdots n_{-3/2}^\downarrow n_{-1/2}^\downarrow \left( n_{1/2}^\downarrow, n_{1/2}^\uparrow \right) \left( n_{3/2}^\downarrow, n_{3/2}^\uparrow \right) \cdots \right\rangle. \tag{7}$$

To summarize, our model is described by a purely interacting Hamiltonian $\mathcal{H}_{\text{int}}$, consisting of the two first Haldane pseudo-potentials, projected to a polarized subspace of the many-body LLL Hilbert space in which no spin up occupies the $n < 0$ orbitals. ED studies shows that some low energy features emerge from the

continuum, as shown in Fig. 1a. The largest reachable system size consists of 4 spin up and 8 spin down particles. For a suitable choice of orbital number, edge excitations acquire a large energy due to finite size effects and we can isolate a single low energy state detached from the continuum. The spin-resolved densities of this vector are depicted in Fig. 1b. They reach plateaus far from the transition, corresponding to the expected results for the Laughlin ($\rho_\uparrow = 0$, $\rho_\downarrow = 1/3$) and Halperin bulks ($\rho_\uparrow = 1/5$, $\rho_\downarrow = 1/5$). It shows that our model Eq. (3) indeed captures the physics of the interface at a microscopic level. The density inhomogeneity persists at the interface and is a probe of the interface reconstruction due to interactions.

**Tensor network description of the bulks.** From the study of quantum entanglement in strongly correlated systems, a new class of variational WFs, namely the tensor networks states (TNS) has emerged in recent years. TNS efficiently encode physically relevant many-body states, relying on their rather low entanglement (for a review, see ref. [45]). For a large set of FQH model states, a MPS–the prototype of TNS–have been derived[46,47]. Moreover, the computational toolbox of tensor networks has been applied to large system size simulations of FQH systems[34,48–53].

We now first briefly recall the theoretical background of the exact MPS description for the Laughlin $1/m$ and the spin singlet Halperin $(m, m, m-1)$ states[34,46,50]. The exact MPS description of an FQH state which can be written as a CFT correlator consists of an electronic part and a background part[46,50]. The former can be deduced from the mode expansion of the primaries appearing in the CFT correlators. The main result of ref. [34] is a method to determine those primaries and the exact MPS representation of two-components Abelian states from a factorization of the $\mathbf{K}$-matrix as $\mathbf{K} = \mathbf{Q}\mathbf{Q}^T$[22,54]. We choose $\mathbf{Q}$ to be upper diagonal for reasons which will become clear later on:

$$\mathbf{K} = \begin{pmatrix} m & m-1 \\ m-1 & m \end{pmatrix}, \quad \mathbf{Q} = \begin{pmatrix} \frac{2m-1}{R_\perp} & \frac{m-1}{R_L} \\ 0 & \frac{m}{R_L} \end{pmatrix} \tag{8}$$

where we have defined $R_\perp = \sqrt{m(2m-1)}$ and $R_L = \sqrt{m}$. The underlying CFT is that of a two-component boson $(\varphi^\perp, \varphi^L)$[55]. Their respective U(1)-charges are integers $n_\perp$, $n_L$ if measured in units of $R_\perp$ and $R_L$ respectively, and satisfy the constraint $\frac{n_\perp + (m-1)n_L}{m} \in \mathbb{Z}$. Compared to ref. [34], Eq. (8) is just a different choice of orthonormal basis for the two-component boson, which

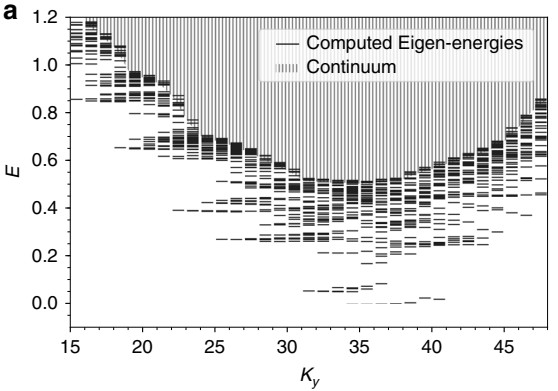

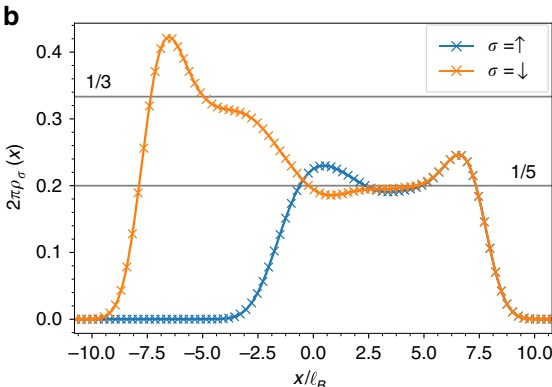

**Fig. 1** Exact diagonalization results. **a** Energy spectrum of Eq. (3) for $N_L = 3$ and $2N_H = 6$ particles in $N_{\text{orb}} = 23$ orbitals, $N_{\text{orb}}^{\text{Lgh}} = 8$ are polarized, on a cylinder of perimeter $L = 12\ell_B$. Low energy features detach from the continuum, hinting toward an interesting low energy physics coming from the interface. **b** Spin-resolved densities for the ground state of Eq. (3) for $N_H = 4$ spin up and $N_H + N_L = 8$ spin down particles in $N_{\text{orb}} = 30$ orbitals, $N_{\text{orb}}^{\text{Lgh}} = 10$ of which are completely polarized. It is the only state detaching from the continuum and arises for a center of mass momentum $K_y = 40.5$. For this system size, the ED calculation captures both the interface and bulk physics as can be seen from the plateaus of densities at both edges of the sample

is related to the usual spin-charge formulation[56,57] by:

$$\varphi^\perp = \sqrt{\frac{1}{2m}}\varphi^c + \sqrt{\frac{2m-1}{2m}}\varphi^s \tag{9a}$$

$$\varphi^L = \sqrt{\frac{2m-1}{2m}}\varphi^c - \sqrt{\frac{1}{2m}}\phi^s, \tag{9b}$$

where $\varphi^c$ and $\varphi^s$ are the bosonic fields corresponding to the charge and spin respectively. We define the spinful electronic operators as

$$\mathcal{W}^\uparrow(z) =: \exp\left(i\sqrt{\frac{2m-1}{m}}\varphi^\perp(z) + i\frac{m-1}{\sqrt{m}}\varphi^L(z)\right) : \chi \tag{10a}$$

$$\mathcal{W}^\downarrow(z) =: \exp\left(i\sqrt{m}\varphi^L(z)\right) : \tag{10b}$$

where $\chi = (-1)^{\frac{n_\perp + (m-1)n_L}{m}}$ acts as a Klein factor ensuring correct commutation relations between the electronic operators. The $j$-th Landau orbital on the cylinder is characterized by its occupation numbers $n^\uparrow$ and $n^\downarrow$. The electronic part $A^{(n^\uparrow, n^\downarrow)}[j]$ of the Halperin MPS matrices only depends on the mode expansion of the electronic operators of Eq. (10):

$$A^{(n^\uparrow, n^\downarrow)}[j] = \frac{1}{\sqrt{n^\uparrow! n^\downarrow!}}\left(\mathcal{W}^\uparrow_{-j}\right)^{n^\uparrow}\left(\mathcal{W}^\downarrow_{-j}\right)^{n^\downarrow} \tag{11}$$

It simply reduces to $A^{(0,n^\downarrow)}[j]$ for the (spin down) polarized Laughlin state. Our variational ansatz relies on the following crucial point: the Laughlin CFT Hilbert space made of a single boson $\varphi^L$ is embedded into the Halperin CFT Hilbert space. Hence both sets of MPS matrices share the same auxiliary space which makes the gluing procedure straightforward in the Landau orbital basis, i.e. a simple matrix multiplication. The basis change of Eq. (9) makes this embedding transparent since we extract $\varphi^L$ from the two component free boson. At the transition between the Halperin and the spin down polarized Laughlin bulks, the cut and glue approach[12] predicts with renormalization group arguments[28] that the degrees of freedom related to $\varphi^L$ gaps out when the tunneling of spin down electrons across the transition is relevant, which is often assumed. The one-dimensional effective field theory at the transition is then expected to be the one of a single bosonic $\varphi^\perp$ field.

To obtain an infinite MPS representation of the states, we combine the electronic operators as $\mathcal{V}(z) = \mathcal{W}^\uparrow(z)|\uparrow\rangle + \mathcal{W}^\downarrow(z)|\downarrow\rangle$. The Operator Product Expansion (OPE) of vertex operators[55] together with the **K**-matrix factorization Eq. (8) ensures that the $N_e$-points correlator

$$\left\langle \mathcal{O}_{bkg}\prod_i \mathcal{V}(z_i)\right\rangle \tag{12}$$

reproduces the Halperin ($m, m, m - 1$) (resp. Laughlin $1/m$) WF with $N_e$ particles through a careful choice of the background charge $\mathcal{O}^H_{bkg}(N_e) = \exp\left\{-iN_e\frac{2m-1}{2}\left(\frac{1}{R_\perp}\varphi_0^\perp + \frac{1}{R_L}\varphi_0^L\right)\right\}$ $\left(\text{resp.}\,\mathcal{O}^L_{bkg}(N_e) = \exp\left\{-iN_e\frac{m}{R_L}\varphi_0^L\right\}\right)$. The choice of the background charge reflects the facts that the Laughlin $1/m$ state is an excitation of the denser Halperin ($m, m, m - 1$) state. Indeed, it may be understood as the introduction of a macroscopic number of $\varphi^\perp$ quasiholes (or a giant quasihole[6]) to fully polarize the Halperin Hall droplet into a Laughlin liquid. Spreading these background charges equally between the orbitals provides a site-independent MPS representation for the Laughlin and Halperin states on the cylinder. Labeling these site-independent MPS representation $B_L$

and $B_H$, we have:

$$B_L^{(n^\downarrow)} = A^{(0,n^\downarrow)}[0]U_L \quad U_L = e^{-\left(\frac{2\pi}{L}\right)^2 L_0^\perp - i\frac{\varphi_0^L}{R_L}}, \tag{13a}$$

$$B_H^{(n^\uparrow, n^\downarrow)} = A^{(n^\uparrow, n^\downarrow)}[0]U_H \quad U_H = e^{-\left(\frac{2\pi}{L}\right)^2 L_0^\perp - i\frac{\varphi_0^\perp}{R_\perp}}U_L, \tag{13b}$$

where $L_0^\perp$ (resp. $L_0^L$) is the Virasoro zero-th mode corresponding to the standard free boson action for $\varphi^\perp$ (resp. $\varphi^L$). Because there is no spin up component in the electronic part of the Laughlin MPS matrices, we may add a shift in the $\varphi^\perp$ U(1)-charge to $U_L$ at each orbital. Doing so allows to always fulfill the compactification constraint $\frac{n_\perp + (m-1)n_L}{m} \in \mathbb{Z}$. Since the Laughlin transfer matrix should only be considered over $m$ orbitals to preserve the topological sectors[50], we simply impose the shift over any $m$ consecutive orbitals to be zero.

**Model wavefunction for the interface.** We introduce a MPS model WF to describe the low energy features of the previously described microscopic model. It has non-vanishing coefficients only over the polarized Hilbert subspace discussed previously. To construct the MPS ansatz, we use $B_H$ (resp. $B_L$) matrices Eq. (13b) (resp. (13a)) for unpolarized (resp. polarized) orbitals. Thus, the MPS ansatz expanded on the many-body states Eq. (7) reads

$$|\Psi_{H-L}\rangle = \sum_{\{n_k^\downarrow\}_k, \{n_k^\uparrow\}_{k>0}} \langle\eta|\cdots B_L^{n_{-3/2}^\downarrow} B_L^{n_{-1/2}^\downarrow} B_H^{\left(n_{1/2}^\uparrow, n_{1/2}^\downarrow\right)} B_H^{\left(n_{1/2}^\uparrow, n_{1/2}^\downarrow\right)}$$
$$\cdots|\mu_\perp\rangle\Big|\{n_k^\downarrow\}_k, \{n_k^\uparrow\}_{k>0}\Big\rangle. \tag{14}$$

This ansatz is schematically depicted on Fig. 2a. Here $\langle\eta|$ and $|\mu\rangle$ are the two states in the auxiliary space fixing the left and right boundary conditions. We fix the gauge of the MPS by choosing a basis for the CFT auxiliary space, which agrees with the structure discussed earlier and in which we have $\langle\eta| = \langle\eta_L| \otimes \langle\eta_\perp|$ and $|\mu\rangle = |\mu_L\rangle \otimes |\mu_\perp\rangle$. It is worth mentioning some important features of this ansatz. First, the use of Halperin and Laughlin site-independent MPS matrices allows to consider an infinite cylinder and enables the use of efficient infinite-MPS (iMPS) algorithms[48,49]. Figure 2b shows the spin-resolved densities of this variational ansatz on an infinite cylinder of perimeter $L = 25\ell_B$. As in Fig. 1b, they smoothly interpolate between the polarized Laughlin bulk at filling factor $\nu_L = 1/3$ and the Halperin unpolarized bulk at filling factor $\nu_H = \frac{1}{5} + \frac{1}{5}$. We recover the typical bulk densities and the spin SU (2) symmetry of the Halperin (332) state after a few magnetic lengths. We can also observe that the finite size effects on the density quickly disappear with increasing $L$. The correlation lengths for the bulks are respectively $\xi_H = 1.28\ell_B$ for the Halperin 332 and $\xi_L = 1.38\ell_B$ for the Laughlin 1/3 states[34]. The ripples disappear for $L/\max(\xi_H, \xi_L) \geq 15$. Another source of finite size effects is the truncation of the infinite CFT Hilbert space in our computations. In practice, we truncate the auxiliary space with respect to the conformal dimension. Our truncation parameter, denoted as $P_{max}$, is a logarithmic measure of the bond dimension (see refs. [34,50]. for a precise definition). The truncation of the auxiliary space is constrained by the entanglement area law[58], the bond dimension should grow exponentially with the cylinder perimeter $L$ to accurately describes the model WFs (at least in the gapped bulks). Thus, as an empirical rule, $P_{max}$ should grow linearly with the cylinder perimeter. This is what prevents us from reaching the thermodynamic limit $L/\ell_B \to \infty$. Using both charge conservation and rotation symmetry along the cylinder perimeter provides additional refinements to the iMPS algorithm[46,48,49]. They can be implemented all along the cylinder, and importantly across the

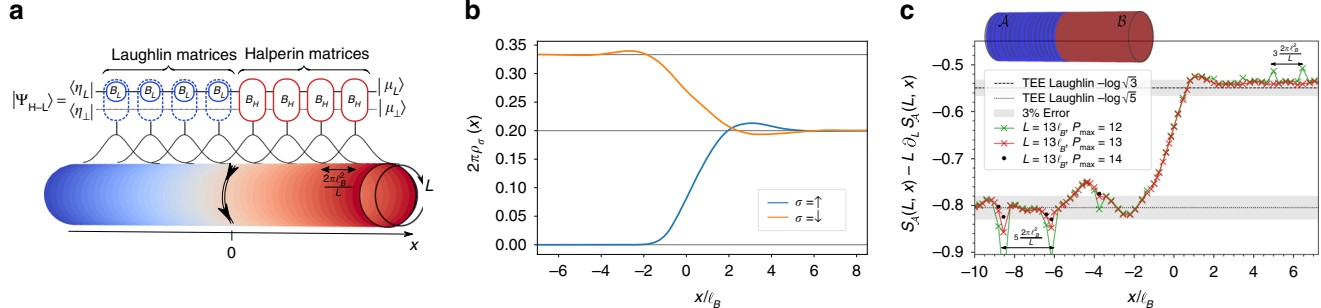

**Fig. 2** Construction of an interface model wavefunction. **a** Schematic representation of the MPS ansatz $|\Psi_{H-L}\rangle$ for the Laughlin 1/3—Halperin 332 interface on a cylinder of perimeter $L$. The Halperin iMPS matrices $B_H$ (red) are glued to the Laughlin iMPS matrices $B_L \otimes \mathbb{I}$ (blue) in the Landau orbital space. Due to the embedding of one auxiliary space into the other, the quantum numbers of $|\mu_\perp\rangle$ (see Eq. (14)) are left unchanged by the Laughlin matrices all the way to the interface. It constitutes a direct access controlling the states of the interface chiral gapless mode, graphically sketched here with a double arrow. **b** Spin-resolved densities of the MPS ansatz state along the cylinder axis obtained at $P_{max} = 11$ for $L = 25\ell_B$. They smoothly interpolate between the Laughlin ($2\pi\rho_\downarrow = 1/3$ and $\rho_\uparrow = 0$) and the Halperin ($2\pi\rho_\downarrow = 2\pi\rho_\uparrow = 1/5$) theoretical values. The density is a robust quantities for which it is safe to consider large perimeter with our truncation level. **c** The EE $S_\mathcal{A}(L, x)$ follows an area law (Eq. 15) for the rotationally invariant bipartition $\mathcal{A} - \mathcal{B}$ depicted on top of the graph. The constant correction is numerically extracted by finite differences $S_\mathcal{A}(L, x) - L\partial_L S_\mathcal{A}(L, x)$ and plotted for $L = 13\ell_B$ as a function of the position along the cylinder axis $x$. It smoothly interpolates between its respective Laughlin and Halperin bulk values and we see no universal signature of the critical mode at the interface. Away from the interface, i.e., $x < -7\ell_B$ on the Halperin side and $x > 3\ell_B$ on the Laughlin side, the extracted $\gamma(x)$ agree with the theoretical expectation within 3% accuracy. Thus, our MPS model WF describes the interface between two distinct topological orders. Note that the extraction and convergence of the subleading quantity $\gamma(x)$ (see Eq. 15) requires large truncation parameters. The spikes appearing on both sides of the transitions are artifacts of the computations of the RSES (see ref. 34 for details). They corresponds to the points where a patch of three (resp. five) orbitals are added on the Laughlin (resp. Halperin) side of the finite size region which translate the bipartition $\mathcal{A} - \mathcal{B}$ to orbital space. They disappear with increasing $P_{max}$, as shown by the points computed at $P_{max} = 14$ (black dots)

interface, by keeping track of the quantum numbers of the CFT states.

Due to the block structure of the MPS matrices[34], the boundary conditions $\langle\eta|$ and $|\mu\rangle$ naturally separates the different charge and momentum sectors all along the cylinder. In particular, they separate the different topological sectors and ensure that no local measurement can discriminate them. For instance on the Halperin side, we can create five distinct bulk WFs (since det $\mathbf{K} = 5$[54]) corresponding to the topological degeneracy of the state on genus one surfaces. While for the Halperin state, the U(1)-charge of both $|\mu_L\rangle$ and $|\mu_\perp\rangle$ should be fixed to determine the topological sector, only the one of $\langle\eta_L|$ is required to fix the topological sector of the Laughlin phase. The remaining bosonic degree of freedom, $\langle\eta_\perp|$ at the edge of the Laughlin bulk constitutes a knob to dial the low-lying excitations of the one dimensional edge mode at the interface. Note indeed that the Laughlin iMPS matrices (15) act as the identity on $\langle\mu_\perp|$ and propagate the state all the way to the interface.

In the following sections, we put our model wavefunctions Eq. (14) to the test and we establish that they indeed capture the universal features of the interface. We confirm that the expected intrinsic topological order is recovered in the bulks away from the interface by extracting the relevant topological entanglement entropies. We characterize the interface gapless mode as a chiral Luttinger liquid. We first extract the interface central charge $c = 1$ through the entanglement entropy. We then extract the corresponding compactification radius $R_\perp = \sqrt{15}$ by identifying the spin and charge of the interface elementary excitations in the many-body spectrum. Moreover, we compare our model states with finite size studies and characterize the low energy features of the spectrum thanks to the boundary state $\langle\eta_\perp|$.

**Universal features of the trial state**. Effective one-dimensional theories similar to the ones of refs. [6,14]. predict that the gapless interface is described by the free bosonic CFT $\varphi^\perp$ of central charge $c = 1$ and compactification radius $R_\perp = \sqrt{15}$. Remarkably, this is neither an edge mode of the Halperin state nor of the

Laughlin state. It is a direct consequence of the edge reconstruction due to interactions which are kept constant across the interface (see Eq. 3). The full characterization of the bulk universal properties and the interface critical theory is the main result of ref. [29]. We briefly discuss such a characterization in the context of the fermionic Laughlin 1/3—Halperin (332) interface.

Local operators such as the density cannot probe the topological content of the bulks. We thus rely on the entanglement entropy (for a review, see ref. [59]) to analyze the topological features of our model WF. All the relevant theoretical framework required for the computation of Real-Space Entanglement Spectrum (RSES)[60–62] has been summed up in the Methods. Consider a bipartition $\mathcal{A} - \mathcal{B}$ of the system defined by a cut perpendicular to the cylinder axis at a position $x$. The RSES and the corresponding Von Neumann EE $S_\mathcal{A}(L, x)$ are computed for various cylinder perimeters $L$. We find that $S_\mathcal{A}(L, x)$ obey an area law[58] for any position of the cut $x$:

$$S_\mathcal{A}(L, x) = \alpha(x)L - \gamma(x). \qquad (15)$$

Far away from the transitions, the constant correction to the area law converges to the Topological Entanglement Entropy (TEE)[63,64] of the Laughlin $\left(\gamma(x \to +\infty) = \log\sqrt{3}\right)$ and Halperin $\left(\gamma(x \to -\infty) = \log\sqrt{5}\right)$ states. Near the interface, the EE still follows Eq. (15) as was recently predicted for such a rotationally invariant bipartition[16]. The correction $\gamma(x)$ smoothly interpolates between its respective Laughlin and Halperin bulk values (see Fig. 2c). Hence, it contains no universal signature of the critical mode at the interface between the two topologically ordered phases. The same conclusion holds for the area law coefficient $\alpha(x)$.

In order to obtain signature from the interface critical theory, we need to break the translation symmetry along the cylinder perimeter. We thus compute the RSES for a bipartition for which the part $\mathcal{A}$ consists of a rectangular patch of length $\ell$ along the compact dimension and width $w$ along the $x$-axis. To fully harness the power of the iMPS approach, it is convenient to add a half-infinite cylinder to the rectangular patch (see Fig. 3a). The contribution of the interface edge mode is isolated with a Levin-

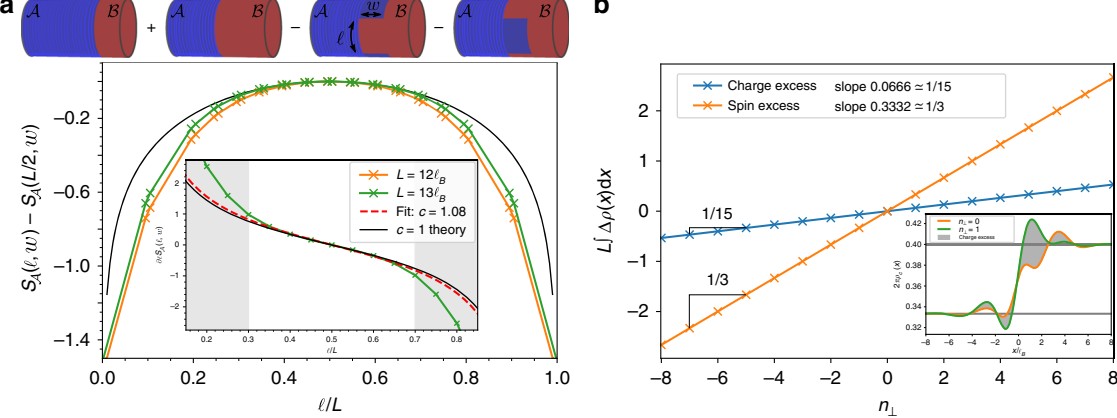

**Fig. 3** Universal features of the interface ansatz. **a** Levin-Wen subtraction scheme (top) to get rid of the spurious area law coming from the patch boundaries along the cylinder perimeter together with corner contributions to the EE. The position and width $w$ of the rectangular extension are selected to fully include the gapless mode at the interface. (bottom) $S_{\mathcal{A}}(\ell, w = 3.25\ell_B)$ for different cylinder perimeters computed at $P_{max} = 12$. They all fall on top of the CFT prediction Eq. (16) with $c = 1$ (black line), pointing toward a critical chiral edge mode at the interface. The agreement improves with increasing $L$, as expected since universal effects are unraveled at $L/\ell_B \to \infty$. The inset shows the derivative $\partial_\ell S_{\mathcal{A}}(\ell, w)$ and the fit with the central charge as the only free parameter (dashed red). The grayed area corresponds to points for which $\ell$ or $L - \ell$ is smaller than three times the Halperin 332 bulk correlation length [37], points in this areas are discarded to mitigate finite-size effects. We find $c_{Fit} = 1.1(1)$, in close agreement with the theoretical prediction (the fitting procedure at $P_{max} = 11$ gives the same result). **b** Charge and spin excess are localized at the interface when excited states are addressed via the MPS boundary U(1)-charge $n_\perp$. Inset shows how to extract the charge excess (gray shaded area) from the charge densities $\rho_c$ at different $n_\perp$, computed at $P_{max} = 11$. Each excess follows a linear relation with respect to $n_\perp$ with extremely good accuracy. The charge (resp. spin) excess has a slope 0.0666(1) $\simeq 1/15$ (resp. 0.3332(2) $\simeq 1/3$). This indicates that the elementary excitations of the $c = 1$ critical theory at the interface carry a fractional charge $e/15$ and a fractional spin 1/3 in unit of the electron spin

Wen addition subtraction scheme[64] depicted in Fig. 3a and more thoroughly discussed in ref. [29]. Noticing that the critical contribution is counted twice, the 1D prediction for a chiral CFT of central charge $c$ with periodic boundary conditions reads[65]

$$S_{\mathcal{A}}(\ell, w) - S_{\mathcal{A}}(L/2, w) = 2\frac{c}{6}\log\left[\sin\left(\frac{\pi\ell}{L}\right)\right].  \quad (16)$$

We vary the length $\ell$ along the compact dimension of the cylinder while keeping $w$ constant, we fit the numerical derivative $\partial_\ell S_{\mathcal{A}}(\ell, w)$ with the theoretical prediction using the central charge $c$ as the only fitting parameter (the derivative removes the area law contribution arising from the cut along $x$). We minimize finite size effects by keeping only the points for which $\ell$ and $L - \ell$ are both greater than three times the Halperin bulk correlation length[34] and consider the largest perimeter that reliably converge $L = 13\ell_B$. We extract a central charge $c = 1.1(1)$ in agreement with the universal expectation. The inset of Fig. 3a shows the numerical data, the result of the fit and the theoretical expectation Eq. (16) which all nicely agree.

In order to fully characterize the gapless mode circulating at the interface, we now extract the charges of its elementary excitations which are related to the compactification radius $R_\perp$. As previously mentioned, excited states of the critical theory are numerically controlled by the U(1)-charge of $\langle \eta_\perp \rangle$. For each of these excited states, we compute the spin-resolved densities and observe that the excess of charge and spin are localized around the interface. They stem from the gapless interface mode observed in Fig. 3a and we plot the charge and spin excess as a function of $n_\perp$ in Fig. 3b. Elementary excitations at the transition carries fractional charge $e/15$ and a fractional spin 1/3 in unit of the electron spin. More generally, the transition between Laughlin $1/m$ state and an Halperin $(m, m, m - 1)$ state should host quasiparticles of charge $e/(m(2m - 1))$. This exactly fits the elementary spin and charge content of $\varphi^\perp$ (see Eq. 9). The interface gapless mode is described

by a chiral Luttinger liquid, i.e. a compact bosonic conformal field theory whose elementary excitations agree with the value $R = \sqrt{15}$ of the compactification radius.

**Comparison with exact diagonalization in finite size**. To go beyond these universal properties and test the relevance of our model WF at a microscopic level, we now compare it to finite size calculations. We first investigate in more details the model WF to get a better microscopic understanding of the interface and the role of the MPS boundary states' quantum numbers. The electronic operators $\mathcal{W}^\uparrow$ and $\mathcal{W}^\downarrow$ generate the charge lattice from a unit cell composed of 5 inequivalent sites[34]. Physically, they correspond to the ground state degeneracy of the Halperin (332) state on the torus (or the infinite cylinder) which is known to be $|\det \mathbf{K}| = 5$[54]. The choice of $n_\perp$ modulo five determines the topological sector of the Halperin bulk far from the transition. An identical analysis involving the spin down electronic operator $\mathcal{W}^\downarrow$ only shows that the Laughlin topological sector is selected by the value of $n_L$ modulo 3. Loosely speaking, these degeneracies give 15 different ways of gluing the two bulks together which lead to the observed fractional charge in Fig. 3b and the compactification radius $R_\perp = \sqrt{15}$. This intuition is rigorous in the thin torus limit $L \ll \ell_B$[67,68] where the bulk physics are dominated by their respective root partitions[42,69]. In the CFT language, we may understand it as a renormalization procedure. Because the Virasoro zero-th mode $L_0$[55] only appears with a prefactor $\left(\frac{2\pi\ell_B}{L}\right)^2$, all excitations above the CFT ground state becomes highly energetic and we can trace them out. This is exactly what the truncation at $P_{max} = 0$ does. From here, we may look at the U(1)-charges $(n_L, n_\perp)$ of the boundary state $|\mu\rangle$ which produce a non-zero coefficient for a given Halperin root partition (i.e., when $P_{max} = 0$). We have performed the study for both the Halperin and the Laughlin bulks, and we summarize our results on the possible ways of gluing together the root partitions of these states in Table 1. These insights on the role of boundary states may be

used to understand the low energy features of finite size calculations. The choice of the U(1)-charges selects one of the possible root configurations given in Table 1. The root configuration fixes the reference for the center of mass angular momentum of the system $K_y$ in finite size. Low energy excitations on top of a given U(1)-charge choice are obtained by dialing the $\varphi^{L}$ and $\varphi^{\perp}$ descendants.

Using the root configuration to relate the MPS boundary indices to the finite size parameters, we are now able to provide convincing numerical evidence that our ansatz should capture the low energy physics of the Hamiltonian Eq. (3). For this purpose, we have performed extensive exact diagonalization (ED) of the Hamiltonian Eq. (3) for $N_L + N_H$ spin down and $N_H$ spin up particles in $N_{orb}^{L} + N_{orb}^{H}$ orbitals, $N_{orb}^{L}$ of which are fully polarized. We would like to show that the low energy features detaching from the continuum in the spectrum of Eq. (3), which is depicted in Fig. 4 for $N_L = N_H = 3$. Let us first fix the level descendant of the MPS boundary conditions $P_\mu = P_\eta = 0$ and selects some U(1)-charges appearing in Table 1, to describe the states which persist in the thin torus limit $L \ll \ell_B$. We observe that, when $N_{orb}^{L} = 3N_L - 2$ and $N_{orb}^{H} = 5N_H$, the ED ground state is the unique state

detaching from the continuum (see green symbols in Fig. 4). It has exactly the total momentum expected from the glued root partition selected by the (0, 0) boundary charges. Our MPS ansatz with these boundary conditions shows extremely high overlap with the corresponding ED ground states (see Table 2). Figure 1b shows the spin-resolved densities of the ED ground state of the largest reachable system sizes. Both the bulks and interface physics are displayed in the ED study and its very high overlap with our ansatz shows that this latest correctly captures the interface physics at a microscopic level.

When the number of polarized orbitals is increased, several low energy branches separate from the continuum (orange and blue markers in Fig. 4). Changing the boundary U(1)-charges of the MPS model WF as prescribed in Table 1 while keeping $P_\mu = P_\eta = 0$, we could identify the root partition dominating the low energy features in each branch in the thin torus limit. These states are labeled by $(n_L, n_\perp)$ in Fig. 4 and their overlaps with the corresponding MPS model WF is always above 0.977. The momentum transfer required to go from one gluing condition to another is extensive with the number of particles. Thus for an

**Table 1 Combining root partitions**

| $(n_L, n_\perp)$ | $[\varnothing\varnothing\uparrow\varnothing\downarrow]$ | $[\varnothing\downarrow\varnothing\varnothing\uparrow]$ | $[\varnothing\uparrow\varnothing\downarrow\varnothing]$ | $[\downarrow\varnothing\varnothing\uparrow\varnothing]$ | $[\uparrow\varnothing\downarrow\varnothing\varnothing]$ |
|---|---|---|---|---|---|
| $[\varnothing\varnothing\downarrow]$ | (0, 0) | (0, 3) | (0, 6) | (0, 9) | (0, 12) |
| $[\varnothing\downarrow\varnothing]$ | (1, −5) | (1, −2) | (1, 1) | (1, 4) | (1, 7) |
| $[\downarrow\varnothing\varnothing]$ | (2, −10) | (2, −7) | (2, −4) | (2, −1) | (2, 2) |

Right and left MPS boundary charges $(n_L, n_\perp)$ to recover the glued Laughlin 1/3 and Halperin 332 root partitions. $\varnothing$, $\downarrow$ and $\uparrow$ respectively denote an empty orbitals or an occupied orbital with a spin down or up. Using the first line and the first columns, the total root configuration should be understood as…$\varnothing$ $\varnothing\downarrow$ $\varnothing\downarrow\varnothing$ $\varnothing\downarrow-\varnothing$ $\varnothing\uparrow\varnothing\downarrow$ $\varnothing\uparrow\varnothing\downarrow$… for the gluing of $[\varnothing \varnothing\downarrow]$ and $[\varnothing \varnothing \uparrow\downarrow]$. Note that defining the root configuration by, e.g., $[\varnothing \varnothing\uparrow\varnothing\downarrow]$ instead of $[\varnothing \varnothing\downarrow\varnothing\uparrow]$ is arbitrary due to the SU(2) singlet nature of the Halperin 332 state. We refer to ref. [66] for more details of the Halperin states root configurations

**Table 2 Comparison with ED ground states**

| $(2N_H, N_L)$ | $\||\psi_{trunc}^{ED}\rangle\|$ | $\||\psi_{trunc}^{MPS}\rangle\|$ | $\langle\psi_{trunc}^{ED}|\psi_{trunc}^{MPS}\rangle$ |
|---|---|---|---|
| (6,3) | 1.000 | 1.000 | 0.998 |
| (6,4) | 1.000 | 1.000 | 0.997 |
| (8,4) | 1.000 | 1.000 | 0.997 |

Overlap between the MPS variational ansatz (at $P_{max} = 11$) for the $(n_L, n_\perp) = (0, 0)$ and $P_\mu = P_\eta = 0$ boundary conditions and the corresponding ED ground state for different system sizes characterized by the particle numbers $(2N_H, N_L)$ on a cylinder $L = 12\ell_B$. The number of orbitals are fixed to $N_{orb}^{L} = 3N_L - 2$ and $N_{orb}^{H} = 5N_H$. Due to the dimension of the many-body Hilbert space considered (415 203 170 for the largest systems), the overlaps are computed over a significant fraction of the vectors weights (we keep all the coefficients with a magnitude greater than $10^{-5}$). The norms of the truncated ED $|\psi_{trunc}^{ED}\rangle$ and MPS $|\psi_{trunc}^{MPS}\rangle$ vectors, which can be evaluated rigorously, give an estimate for the possible error

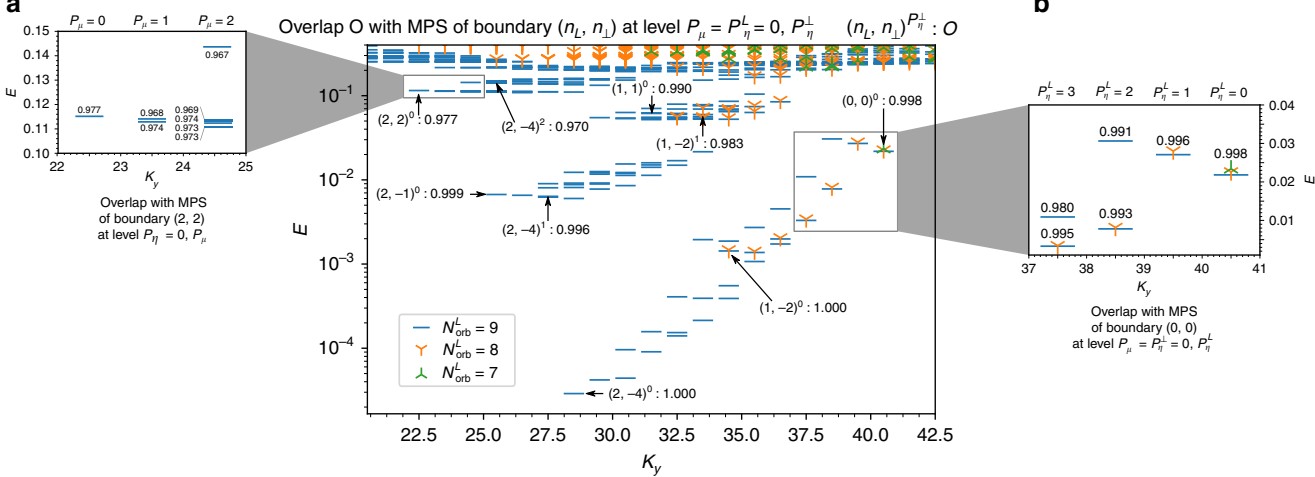

**Fig. 4** Identifying low-energy excitations. Energy spectrum (using a logarithmic scale) of Eq. (3) for a system of 9 particles (6 spin down, 3 spin up) on a cylinder of perimeter $L = 12\ell_B$ with $N_{orb}^{H} = 15$ orbitals and $N_{orb}^{L} = 7$ (green symbols), $N_{orb}^{L} = 8$ (orange symbols) or $N_{orb}^{L} = 9$ (blue lines). In the first case, only one of the root configuration of Table 1 can be produced. Increasing the number of polarized orbitals, we allow for excitations of the Laughlin edge gapless mode, that we can track with the MPS boundary quantum number $P_\eta^{L}$ (**a**). The low lying branches for $N_{orb}^{L} = 8$ or 9 are all connected to some gluing conditions (see Table 1). We have indicated them as $(n_L, n_\perp)^0$, together with the overlap with the ED states targeted (Main Figure). Halperin edge excitations are also present in the spectrum and, as shown in **b**, we can locate them in the spectrum thanks to the MPS ansatz by varying $P_\mu$. Finally, we can discriminate the interface gapless mode excited states by comparing the ED eigenvectors with our MPS model WF when $P_\eta^{\perp} \neq 0$. The corresponding states and overlaps are labeled $(n_L, n_\perp)^{P_\eta^{\perp}}$. We can for instance follow the two first excitations starting from $(2, -4)^0$, and observe a very steep dispersion relation for the interface critical mode. In general, all low lying excitations may be reproduced with our ansatz for mixed excitation ($P_\mu \neq 0$, $P_\eta^{L} \neq 0$, $P_\eta^{\perp} \neq 0$). All the overlaps presented were computed at truncation parameter $P_{max} = 12$ and as a rule of thumb we observe that the closer to the continuum the poorer the MPS ansatz performs

infinite system, the different branches clearly separate in the spectrum, while they may overlap (and do in many cases) for the system considered as can be seen for the state (1, 1) in Fig. 4. Furthermore, we are able to access and identify the low energy excitations above the different root configurations with the MPS model ansatz. The latter arises at the edge of the gapped FQH droplets and are of three distinct types: Laughlin edge excitations, Halperin edge excitations and excitations of the interface gapless mode. We now exemplify each of these three cases, and show at the same time how to characterize the states in the many-body spectrum.

*Halperin edge excitations*: Let us fix the MPS boundary charges to $(n_L, n_\perp) = (2, 2)$, whose corresponding root partition (see Table 1) is expected to appear for a center of mass momentum $K_y = 22.5$. In this momentum sector, we find one eigenvector of Eq. (3) with high overlap with the model WF for $P_\mu = P_\eta = 0$, which slightly detaches from the continuum. Changing the momentum of the Halperin boundary state $P_\mu$, the MPS model WFs acquire a momentum $K_y + P_\mu$. These excitations are used to describe quasihole excitations at the edge of the Halperin bulks[66,69]. We compute the weight of the ED eigenvectors close-by in the space generated by all MPS model WFs with $P_\eta = 0$ and $P_\mu = 1$ or $P_\mu = 2$. As shown in Fig. 4a, they clearly belong to this subspace. Thus, our MPS model WFs allowed us to characterize without ambiguity the excitations in the branch starting at $K_y = 22.5$ as Halperin gapless edge mode excitations (note the counting 1-2-5-…).

*Laughlin edge excitations*: The same tools may be used to probe excitations of the Laughlin gapless edge mode. We split $P_\eta = P_\eta^L + P_\eta^\perp$, where $P_\eta^\perp$ (resp. $P_\eta^L$) denotes the descendant level of the state $\langle \eta |$ with respect to the $\varphi^\perp$ (resp. $\varphi^L$) boson and we keep $P_\eta^\perp = 0$ for now. Starting from the state $(n_L, n_\perp) = (0, 0)$ at $K_y = 40.5$, we could reproduce the excitations at $K_y - P_\eta^L$ as depicted in Fig. 4b. Finite size effects limit the number of accessible descendants to $P_\eta^L = 3$ in the ED spectrum. However, it is clear from the computed overlaps that the considered states are edge excitations of the Laughlin droplet. The same analysis can be repeated all over the spectrum.

*Interface excitations*: Finally, and more interestingly, we were able to localize the excitations due to the interface gapless mode (see Fig. 4). This time, we keep $P_\mu = P_\eta^L = 0$ and vary $P_\eta^\perp$. We really want to highlight the difficulty to find those states from a pure ED approach, especially considering that each interface gapless mode excitation changes the energy by an order of magnitude. We attribute this large interface mode velocity to the sharpness of the transition described by our ansatz, i.e., to the change from $\mu_\uparrow = 0$ to $\mu_\uparrow = \infty$ over an inter-orbital distance (see Supplementary Fig. 2 for further investigations).

While we have only considered one kind of excitations at a time, generic low energy states in the spectrum are characterized by non-zero $P_\mu$, $P_\eta^L$ and $P_\eta^\perp$. While the ED spectrum does not distinguish between the Laughlin and Halperin bulk excitations and the excitations of interface modes, the high overlap between the MPS states with the low lying part of the ED spectra help us discriminating these different types of excitations. This makes the proposed model states valuable tools even for finite size studies.

## Discussion

We have considered the fermionic interface between the Laughlin 1/3 and Halperin (332) states, relevant for condensed matter experiments. Indeed experimental realizations of this transition can be envisioned in graphene. There, the valley degeneracy leads to a spin singlet state at $\nu = 2/5$[30,31] while the system at $\nu = 1/3$ is

spontaneously valley-polarized[31–33]. Thus, changing the density through a top gate provides a direct implementation of our setup.

In ref. [29], we introduced a family of model states to describe the Laughlin–Halperin interface. Their universal properties were established using quantum entanglement measures, and the emerging gapless mode at the interface was characterized. It is described by a chiral Luttinger liquid, i.e., a compact bosonic conformal field theory whose elementary excitations agree with the value $R = \sqrt{15}$ of the compactification radius.

The main result of this work is the thorough microscopic validation of our model wavefunctions. We introduced an experimentally relevant microscopic Hamiltonian that captures the physics of this interface, which we then analysed using exact diagonalization simulations on large-size systems. We found that the family of model states we introduced in ref. [29] performs exceedingly well, reproducing the low-energy states of the microscopic model with extraordinarily good overlaps. Furthermore, these model states provide a powerful tool to identify the nature of the low-energy states obtained through exact diagonalization. In particular, they allow to disentangle the interface modes from the Laughlin and Halperin edge modes, a notoriously difficult task to carry out from finite size exact diagonalization.

Our interface model state, therefore, yields a bridge between the microscopic, experimentally relevant model and its low-energy effective description in terms of interfaces between topological quantum field theories.

## Methods

**Entanglement entropy and MPS: derivation**. We turn to the computation of the real space entanglement spectrum (RSES)[60–62] for MPS ansatz considered. We recall that the electronic operator modes have the same commutation relations as the creation and annihilation operators. We will use these relations extensively to compute the RSES. For clarity, we focus primarily on the Laughlin case, the generalization to the Halperin case or to the Laughlin–Halperin interface only involves additional indices without involving any new technical step. For clarity, we will remove the spin index from the discussion whenever they are not needed. It is also useful to work with the site-dependent representation of the Laughlin $1/m$ state[50]:

$$\left| \Phi_{\alpha_R}^{\alpha_L} \right\rangle = \sum_{\{n_k^\perp\}} \langle \alpha_L | \mathcal{O}_{bkg}^L \left[ \prod_{k=0}^{N_\varphi} \frac{1}{n_k^\perp!} \left( \mathcal{W}_{-k}^\perp \right)^{n_k^\perp} \otimes \left( c_k^\dagger \right)^{n_k^\perp} \right] (|\alpha_R\rangle \otimes |\Omega\rangle), \quad (17)$$

where $c_k^\dagger$ creates a particle on orbital $k$ (see Eq. (4)). The afore-mentioned site independent MPS representation Eq. (13a) comes from spreading of the background charge[46] and requires a shift of the MPS-boundary state U(1) charges[50], which we will keep implicit here.

*Real Space Bipartition*—Under a generic real space bipartition $\mathcal{A} - \mathcal{B}$, the LLL orbitals Eq. (4) are decomposed as $c_k^\dagger = d_{k,\mathcal{A}}^\dagger + d_{k,\mathcal{B}}^\dagger$ with

$$d_{k,\mathcal{I}}^\dagger = \int_{r\in\mathcal{I}} d^2\mathbf{r}\,\psi_k(\mathbf{r})c_k^\dagger(\mathbf{r}), \quad \mathcal{I} \in \{\mathcal{A}, \mathcal{B}\}. \quad (18)$$

The sets $\{d_{k,\mathcal{A}}\}$ and $\{d_{k,\mathcal{B}}\}$ span two disjoint Hilbert spaces of respective vacua $|\Omega_\mathcal{A}\rangle$ and $|\Omega_\mathcal{B}\rangle$ but are in general not orthonormal:

$$\left\{ d_{k,\mathcal{I}}, d_{\ell,\mathcal{I}'}^\dagger \right\} = \delta_{\mathcal{I},\mathcal{I}'} \int_{r\in\mathcal{I}} d^2\mathbf{r}\,\psi_k^*(\mathbf{r})\psi_\ell(\mathbf{r}) \quad (19)$$

where $\{,\}$ denotes the anticommutator. For a cut preserving the rotation symmetry along the cylinder perimeter $\mathcal{A} = \{(x',y')|x' < x, 0 \le y' \le L\}$, the overlaps in the right-hand side of Eq. (19) are diagonal and take the form

$$g_{k,\mathcal{A}} = \int_{r\in\mathcal{A}} d^2\mathbf{r}\,\psi_k^*(\mathbf{r})\psi_k(\mathbf{r})$$
$$= \sqrt{\frac{1}{\pi\ell_B}\int_{x'<x}dx'\exp\left(-\frac{(x'-x_k)^2}{\ell_B^2}\right)} \quad (20)$$

These overlaps can still be computed analytically for some bipartitions breaking the rotation symmetry. In that case, $\{d_{k,\mathcal{A}}\}$ can be decomposed over an orthonormal basis $\{\bar{c}_{\mu,\mathcal{A}}\}$ as

$$d_{k,\mathcal{A}}^\dagger = \sum_{\mu=0}^{N_\phi} \alpha_{k,\mu}\bar{c}_{\mu,\mathcal{A}}^\dagger \quad (21)$$

where the coefficient $\{\alpha_{k,\mu}\}$ are obtained either analytically or numerically from the known overlaps between LLL orbitals over the region $\mathcal{A}$.

*Split and Swap Procedure*—Using the decomposition $c_{k,\downarrow}^\dagger = d_{k,\mathcal{A}}^\dagger + d_{k,\mathcal{B}}^\dagger$ together with the commutation relations of Eq. (19), and introducing a closure relation

$\sum_{\beta \in \mathcal{H}_{\mathrm{CFT}}} |\beta\rangle\langle\beta|$, with $\mathcal{H}_{\mathrm{CFT}}$ the auxiliary space, i.e., the CFT Hilbert space, the Schmidt decomposition of Eq. (17) $\left|\Phi^{\alpha_L}_{\alpha_R}\right\rangle = \sum_{\beta \in \mathcal{H}_{\mathrm{CFT}}} |\phi^B_\beta\rangle \otimes |\phi^A_\beta\rangle$ onto the partition $\mathcal{A} - \mathcal{B}$ is found to be:

$$\left|\phi^B_\beta\right\rangle = \sum_{\{n^\downarrow_k\}} \langle\alpha_L| \mathcal{O}^L_{\mathrm{bkg}} \left[\prod_{k=0}^{N_\phi} \frac{1}{n^\downarrow_k!} \left(\mathcal{W}^\downarrow_{-k}\right)^{n^\downarrow_k} \otimes \left(c^\dagger_k\right)^{n^\downarrow_k}\right] (|\alpha_R\rangle \otimes |\Omega\rangle),$$
$$\left|\phi^B_\beta\right\rangle = \sum_{\{n^\downarrow_k\}} \langle\tilde\alpha_L| \left[\prod_{k=0}^{N_\phi} \frac{1}{n^\downarrow_k!} \left(\mathcal{W}^\downarrow_{-k}\right)^{n^\downarrow_k} \otimes \left(c^\dagger_k\right)^{n^\downarrow_k}\right] (|\alpha_R\rangle \otimes |\Omega\rangle),$$

(22)

where we have set $\langle\tilde\alpha_L| = \langle\alpha_L| \mathcal{O}^L_{\mathrm{bkg}}$. This step is described in great details for the rotationally symmetric case in ref. [34,46]. From now on, we focus on subspace $\mathcal{A}$, the derivation being exactly the same for the subspace $\mathcal{B}$. The occupation numbers $\{n^A_k\}$ are equivalently described by ordered lists of occupied orbitals $\lambda = (\lambda_1, \cdots, \lambda_{N_e})$, with $N_e$ the number of electrons in the system: $N_\phi \geq \lambda_1 > \cdots > \lambda_{N_e} \geq 0$. Because of the commutation relation of the vertex operator modes, we may also write

$$\left|\phi^A_\beta\right\rangle = \sum_{\lambda_1, \cdots, \lambda_{N_e}} \langle\beta| \left[\prod_{j=1}^{N_e} \mathcal{W}^\downarrow_{-\lambda_j} \otimes d^\dagger_{\lambda_j, \mathcal{A}}\right] (|\alpha_R\rangle \otimes |\Omega_\mathcal{A}\rangle), \tag{23}$$

where the sum runs over unordered lists of integers $\{\lambda\}$. Plugging the orthonormal basis with Eq. (21) and reordering the various terms, we find

$$\left|\phi^A_\beta\right\rangle = \sum_{\mu_1, \cdots, \mu_{N_e}} \langle\beta| \left[\prod_{j=0}^{N_\phi} \left(\sum_{\lambda=0}^{N_\phi} \alpha_{\lambda,\mu_j} \mathcal{W}^\downarrow_{-\lambda}\right) \otimes \tilde c^\dagger_{\mu_j, \mathcal{A}}\right] (|\alpha_R\rangle \otimes |0_\mathcal{A}\rangle). \tag{24}$$

A similar reasoning helps us to finally expressing the state $|\phi^A_\beta\rangle$ in the occupation basis $\tilde n^A_k$ relative to the new physical space spanned by the orthonormal basis $\{\tilde c_{k,\mathcal{A}}\}$. We find the MPS expression

$$\left|\phi^A_\beta\right\rangle = \sum_{\{\tilde n^A_k\}} \langle\beta| K^{\tilde n^A_{N_\phi}}_\mathcal{A}[N_\phi] \cdots K^{\tilde n^A_0}_\mathcal{A}[0] |\alpha_R\rangle |\tilde n^A_{N_\phi} \cdots \tilde n^A_0\rangle, \qquad K^{\tilde n}_\mathcal{A}[j] = \frac{1}{\sqrt{\tilde n!}} \left(\sum_{\lambda=0}^{N_\phi} \alpha_{\lambda,j} \mathcal{W}^\downarrow_{-\lambda}\right)^{\tilde n}. \tag{25}$$

Here, we have used the commutation relations of the vertex operator modes to swap the matrices in order to derive Eq. (25), a site-dependent representation of $|\phi^A_\beta\rangle$ onto the orthonormal basis $\{\tilde c_{k,\mathcal{A}}\}$.

*Spreading the Background Charge*—The last step of the derivation consists in spreading the background charge in order to find back the iMPS matrices Eq. (13a) far away from the cut. The Laughlin background charge $\mathcal{O}^L_{\mathrm{bkg}}$ can only be spread over $N_{\mathrm{orb}}$ orbitals, but the bipartition has introduced twice more matrices ($N_{\mathrm{orb}}$ in both parts $\mathcal{A}$ and $\mathcal{B}$). Although any allocation of the background charge over these matrices is acceptable, we append $U_L$ to the first (resp. last) $N_{\mathrm{orb}}/2$ matrices of $\mathcal{A}$ (resp. $\mathcal{B}$). The product of matrices appearing in Eq. (25) can be split into two parts

$$\left(F^{\tilde n^A_{N_\phi}}_\mathcal{A}\left[N_\phi, N_{\mathrm{orb}}/2\right] \cdots F^{\tilde n^A_{N_{\mathrm{orb}}/2}}_\mathcal{A}\left[N_{\mathrm{orb}}/2, 1\right]\right) \left(F^{\tilde n^A_{N_{\mathrm{orb}}/2-1}}_\mathcal{A}[N_{\mathrm{orb}}/2 - 1, 0] U_L \cdots F^{\tilde n^A_0}_\mathcal{A}[0, 0] U_L\right)$$

(26)

where we have defined

$$F^{\tilde n}_\mathcal{A}[j, q] = \frac{1}{\sqrt{\tilde n!}} \left(\sum_{\lambda=0}^{N_\phi} \alpha_{\lambda,j} \mathcal{W}^\downarrow_{-(\lambda-j)+q}\right)^{\tilde n}. \tag{27}$$

For a partition preserving the rotation symmetry around the cylinder axis, we have $\alpha_{k,r} = \delta_{k,r} g_{k,\mathcal{A}}$ (see Eq. 20) and we thus recover the tensor of refs. [34,46,50]. For the rectangular patch described in the main text, the off diagonal weights $\alpha_{k,r}$ decay rapidly for orbitals far from the cut (typically like Gaussian factors multiplied by cardinal sine functions) so that we can approximate $\alpha_{k,r} \simeq \delta_{k,r} g_{k,\mathcal{A}}$. In other words, the rotational symmetry is recovered after a large enough number of orbitals. Moreover, far away from the cut in the iMPS part of the product, $g_{k,\mathcal{A}} = 1$ and we get back the site independent matrices $F^{(\tilde n)}_\mathcal{A}[k, 0] U_L = B^{(\tilde n)}_L$. Similarly when $g_{k,\mathcal{A}} = 0$, i.e. far away from the cut in the site-dependent part of the product, the matrices reduces to $F^{\tilde n}_\mathcal{A}[j, q] \simeq \delta_{\tilde n, 0} \mathbf{1}$, with $\mathbf{1}$ being the identity operator over the auxiliary space. This shows that the translation invariance along the cylinder axis is recovered far away from the transition. We can thus work on the infinite cylinder and take $N_{\mathrm{orb}} \to \infty$, by switching to the site independent matrices far away from the cut. Numerically, we have considered up to 50 orbitals in the site-dependent region to ensure that when the iMPS is glued to take the limit $N_{\mathrm{orb}} \to +\infty$, we always satisfy the condition $|\alpha_{k,r} - \delta_{k,r}| < 10^{-10}$.

## Data availability

Raw data and additional results supporting the findings of this study are included in Supplementary Information and are available from the corresponding author on request. The exact diagonalization data for finite size comparisons have been generated using the software "DiagHam" (under the GPL license).

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

## Acknowledgements

We thank E. Fradkin, J. Dubail, A. Stern and M.O. Goerbig for enlightening discussions. We are also grateful to B.A. Bernevig and P. Lecheminant for useful comments and collaboration on previous works. V.C., B.E. and N.R. were supported by the grant ANR TNSTRONG No. ANR-16-CE30-0025 and ANR TopO No. ANR-17-CE30-0013-01.

## Author contributions

V.C., N.C. and N.R. developed the numerical code. V.C., N.C. and N.R. performed the numerical calculations. V.C., N.C., N.R. and B.E. contributed to the analysis of the numerical data and to the writing of the manuscript.

## Additional information

**Competing interests:** The authors declare no competing interests.

