## [Peer Review File · Nature Communications]

REVIEWERS' COMMENTS:

Reviewer #1 (Remarks to the Author):

Dear Authors and Dear Editors of Nature Communications,

First of all, please, once again, excuse me for the long delay in producing this review. Earlier, I reviewed the paper "A variational approach to chiral topological order interfaces", by the same authors. I considered this to be excellent work and recommended it for publication, although I had some reservations about the form of the paper. In particular, I noted that much material that could easily qualify for the main paper had been relegated to the supplementary material, notably the direct exact diagonalization study of the interface between two chiral, fermionic, topological phases which had a decent chance to be experimentally realized.

In the current paper, "Microscopic Study of the Halperin-Laughlin Interface through Matrix Product States", the authors have elevated that study, combined with several other parts of the supplementary data to the original manuscript, to a self-contained paper. The current paper is important in several ways. First of all, it provides direct confirmation that the trial wave functions proposed in the original manuscript, and now in the companion paper "Model States for a Class of Chiral Topological Order Interfaces", give an excellent description of the relevant systems at small system sizes. It is a little unfortunate that the bosonic trial states directly mentioned in the companion paper are treated in the supplementary information here, but it is understandable from the point of view of the current manuscript. The paper is also important as it deals with a situation that could be experimentally realized, for example in graphene, as the authors indicate. This is an exciting prospect, because, while interfaces between chiral topological phases in two dimensions do occur in quantum Hall experiments, they are usually hard to probe and more importantly hard to handle theoretically. The combination of the authors' trial wave function technology with state of the art numerical techniques allows for a more detailed understanding of this particular system (as well as others like it) and could stimulate and guide experimental efforts in the near future.

In view of the above, I recommend this paper for publication.

I do have a number of remarks which I would like the authors to address.

1. First of all, while the authors propose a very interesting potential experiment in graphene, they don't make much effort to indicate how their theoretical framework could be adapted to more closely mimic the experimental situation. It would be worthwhile to add some remarks on this. For example, the edge at the interface in a real system will likely be far "softer", but this could be taken into account with modification of the trial wave functions, breaking the translational symmetry of the ansatz explicitly near the interface

2. The conclusion of the paper states that the paper validates that the trial wave functions give a good description of the low energy spectrum of an "Experimentally relevant microscopic Hamiltonian". While it is almost certainly true that the Hamiltonian used captures universal features of the interface, it is probably also fair to say that this Hamiltonian is not a realistic description of even an idealized experiment. Rather it is a model Hamiltonian which on each side of the interface would give precisely the local bulk trial state there as its ground state. Of course it is well known that ground states for this type of Hamiltonian are often very close to those for more realistic interactions, but in the excited states, even at low energy, there can often be very important corrections when longer range interaction terms are introduced. Especially in fermionic systems, such terms are likely important. Symmetry breaking terms could also appear in graphene due to the presence of the atomic lattice. It

would be interesting to know if any of the detailed structure observed in the excitation spectrum of the Hamiltonian used here would survive the introduction of such long range and/or symmetry breaking interaction terms.

Some further very minor points and typos:

3. Pg. 2 (Introduction)

"The main results ... appears"

Should be "result appears" or "results appear"

4. Fig. 1 caption "arise" Should be "arises",
also "system sizes" should be "system size"

5. below formula (2)

"total many body state particles" remove "particles"

"vanishing properties... ensures" -> ensure

A reference to a paper discussing vanishing properties might be appropriate here.

6. After Eq. (7) it says "a single low energy state detaches from the continuum with a center of mass momentum $K_y=68$ (...) depicted in Fig. 1b."

However, in the figure, $K_y=40.5$.

Also, it is not clear to me from the text whether this state is the single state emerging from the vacuum at this value of K_y (as it appears from Fig. 1a) or whether it is the single state to emerge from the vacuum at all (The latter possibility actually appears to be the case when looking at Fig. 6)

7. Below Eq. (11)

"the degrees of freedom related to ϕ^L should gap out".

This refers to the theoretical picture where the states have independent edges which interact ("cut and glue" formalism). Maybe clarify a little.

8. Below Eq. (13)

"Since the Laughlin transfer matrix should only be considered over m orbitals ... we simply impose that this shift in n_{perp} be zero every m orbital"

It seems to be intended that total of the shifts of any m consecutive orbitals is zero?

9. pg. 7

Note that compared to the bosonic case discussed in Ref. [29] ...disappear typically for $L/\max(\xi_h, \xi_l) \geq 15 l_b$

There appears to be no comparison?

Also, the formula should not have l_b on the RHS.

10. Supplementary Table IV "Many body Hilbert" - needs "space"

11. last paragraph, supplement

"bound dimension" -> "bond dimension"

"large corrections to the this linear behavior" (remove "the")

Reviewer 1 (Remarks to the Author):

« Dear Authors and Dear Editors of Nature Communications,

First of all, please, once again, excuse me for the long delay in producing this review. Earlier, I reviewed the paper "A variational approach to chiral topological order interfaces", by the same authors. I considered this to be excellent work and recommended it for publication, although I had some reservations about the form of the paper. In particular, I noted that much material that could easily qualify for the main paper had been relegated to the supplementary material, notably the direct exact diagonalization study of the interface between two chiral, fermionic, topological phases which had a decent chance to be experimentally realized.

In the current paper, "Microscopic Study of the Halperin-Laughlin Interface through Matrix Product States", the authors have elevated that study, combined with several other parts of the supplementary data to the original manuscript, to a self-contained paper. The current paper is important in several ways. First of all, it provides direct confirmation that the trial wave functions proposed in the original manuscript, and now in the companion paper "Model States for a Class of Chiral Topological Order Interfaces", give an excellent description of the relevant systems at small system sizes. It is a little unfortunate that the bosonic trial states directly mentioned in the companion paper are treated in the supplementary information here, but it is understandable from the point of view of the current manuscript. The paper is also important as it deals with a situation that could be experimentally realized, for example in graphene, as the authors indicate. This is an exciting prospect, because, while interfaces between chiral topological phases in two dimensions do occur in quantum Hall experiments, they are usually hard to probe and more importantly hard to handle theoretically. The combination of the authors' trial wave function technology with state of the art numerical techniques allows for a more detailed understanding of this particular system (as well as others like it) and could stimulate and guide experimental efforts in the near future.

In view of the above, I recommend this paper for publication.

I do have a number of remarks which I would like the authors to address.

1. First of all, while the authors propose a very interesting potential experiment in graphene, they don't make much effort to indicate how their theoretical framework could be adapted to more closely mimic the experimental situation. It would be worthwhile to add some remarks on this. For example, the edge at the interface in a real system will likely be far "softer", but this could be taken into account with modification of the trial wave functions, breaking the translational symmetry of the ansatz explicitly near the interface.

2. The conclusion of the paper states that the paper validates that the trial wave functions give a good description of the low energy spectrum of an "Experimentally relevant microscopic Hamiltonian". While it is almost certainly true that the Hamiltonian used captures universal features of the interface, it is probably also fair to say that this Hamiltonian is not a realistic description of even an idealized experiment. Rather it is a model Hamiltonian which on each side of the interface would give precisely the local bulk trial state there as its ground state. Of course it is well known that ground states for this type of Hamiltonian are often very close to those for more realistic interactions, but in the excited states, even at low energy, there can often be very important corrections when longer range interaction terms are introduced. Especially in fermionic systems, such terms are likely important. Symmetry breaking terms could also appear in graphene due to the presence of the atomic lattice. It would be interesting to know if any of the detailed structure observed in the excitation spectrum of the Hamiltonian used here would survive the introduction of such long range and/or symmetry breaking interaction terms.

Some further very minor points and typos:

3. Pg. 2 (Introduction) "The main results ... appears" Should be "result appears" or "results appear"

4. Fig. 1 caption "arise" Should be "arises", also "system sizes" should be "system size"

5. below formula (2) "total many body state particles" remove "particles" "vanishing properties... ensures" -> ensure A reference to a paper discussing vanishing properties might be appropriate here.

6. After Eq. (7) it says "a single low energy state detaches from the continuum with a center of mass momentum $K_y = 68$ (...) depicted in Fig. 1b." However, in the figure, $K_y = 40.5$. Also, it is not clear to me from the text whether this state is the single state emerging from the vacuum at this value of K_y (as it appears from Fig. 1a) or whether it is the single state to emerge from the vacuum at all (The latter possibility actually appears to be the case when looking at Fig. 6)

7. Below Eq. (11) "the degrees of freedom related to ϕ^L should gap out". This refers to the theoretical picture where the states have independent edges which interact ("cut and glue" formalism). Maybe clarify a little.

8. Below Eq. (13) "Since the Laughlin transfer matrix should only be considered over m orbitals ... we simply impose that this shift in n_{\perp} be zero every m orbital" It seems to be intended that total of the shifts of any m consecutive orbitals is zero?

9. pg. 7 Note that compared to the bosonic case discussed in Ref. [29] ...disappear typically for $L/\max(\xi_H, \xi_L) \geq 15\ell_B$ There appears to be no comparison? Also, the formula should not have ℓ_B on the RHS.

10. Supplementary Table IV "Many body Hilbert" - needs "space"

11. last paragraph, supplement "bound dimension" -> "bond dimension" "large corrections to the this linear behavior" (remove "the") »

Reply to Reviewer 1:

We acknowledge Reviewer 1 once again for the three detailed and insightful reports she/he issued during the whole referral process.

Her/His remarks 3 to 11 present minor points and typos that we have clarified in the main text. We thank her/him for highlighting them. The most important clarification that we implemented was about Fig.1a and Fig.1b that were not obtain for the same system sizes. Though written in the caption of Fig.1, we understand the potential confusion when reading the text. We have added a sentence to make this point clearer.

Remarks 1 and 2 require a more detailed discussion:

1. Our ansatz describes a sharp interface which, as Reviewer 1 points out, will likely be softer in real experiments. Note that our interface is already smooth in real space (but sharp in orbital space). We put many thoughts in the generalization of our ansatz to a softer transition between the two topological phases. Analytically, such a generalization may be implemented by weighting the MPS matrices close to the interface with multiplicative factors as derived in the Methods section for the RSES. The reason why we omitted this discussion in the text is purely technical as most of the numerical computations become untractable when introducing these weights. For instance, the extraction of the central charge requires much more orbitals (see Methods) and denser matrices on the Laughlin side, making the computation extremely time-consuming. The comparison with Exact Diagonalization (ED) would also become more complicated: it is already difficult to reach the bulk physics on either side of the sharp transition in ED. A smoother transition would require an even bigger system. Although the softness of the interface is an important question to make our prediction even more quantitative, technical limitations prevented us to investigated it systematically.
2. The second remark deals with the short interaction range used in the manuscript which do not fully account for the Coulombic interaction in electronic systems. Though the spectrum (especially the interface excitations) may rearrange, we expect the spin or charge nature of the excitations to persist. While the Reviewer agrees that the literature on the subject justifies our approach for the ground state properties in both bulks, she/he raises the question of the excited states. The methodology to answer this problem is far beyond the scope of the present manuscript. Indeed, the analytical description of the transition pointed out in the paper should be abandoned (no equivalent results exist for true long range interaction) and variational optimization methods should probably used. One could imagine using iDMRG to obtain infinite boundary conditions for the Laughlin and Halperin phases and optimize a finite region of tensors (finite size DMRG) around the transition. This would require several of technological development and is far out of the scope of our paper.

The presence of the atomic lattice is indeed another physically interesting question. We would like to point out the reference [Hunt et al. Nat. Comm. Vol. 8, 948 (2017)] which shows how to apply our techniques for lattice system. It is definitely something worth looking at in the future but once again requires a lot of development.